# *Epichloë* Endophyte Infection Rates and Alkaloid Content in Commercially Available Grass Seed Mixtures in Europe

**DOI:** 10.3390/microorganisms8040498

**Published:** 2020-03-31

**Authors:** Jochen Krauss, Veronika Vikuk, Carolyn A. Young, Markus Krischke, Martin J. Mueller, Katja Baerenfaller

**Affiliations:** 1Department of Animal Ecology and Tropical Biology, University of Würzburg, 97074 Würzburg, Germany; veronika.vikuk@uni-wuerzburg.de; 2Noble Research Institute LLC, Ardmore, OK 73401, USA; cayoung@noble.org; 3Department of Pharmaceutical Biology, Metabolomics Core Unit, University of Würzburg, 97082 Würzburg, Germany, martin.mueller@biozentrum.uni-wuerzburg.de (M.J.M.); 4Swiss Institute of Allergy and Asthma Research (SIAF), University of Zurich, and Swiss Institute of Bioinformatics (SIB), 7265 Davos, Switzerland; katja.baerenfaller@siaf.uzh.ch

**Keywords:** *Epichloë* spp., grass endophytes, cool-season grass species, infection rates, alkaloids, toxicity, livestock, horses, *Lolium perenne*, perennial ryegrass

## Abstract

Fungal endophytes of the genus *Epichloë* live symbiotically in cool season grass species and can produce alkaloids toxic to insects and vertebrates, yet reports of intoxication of grazing animals have been rare in Europe in contrast to overseas. However, due to the beneficial resistance traits observed in *Epichloë* infected grasses, the inclusion of *Epichloë* in seed mixtures might become increasingly advantageous. Despite the toxicity of fungal alkaloids, European seed mixtures are rarely tested for *Epichloë* infection and their infection status is unknown for consumers. In this study, we tested 24 commercially available seed mixtures for their infection rates with *Epichloë* endophytes and measured the concentrations of the alkaloids ergovaline, lolitrem B, paxilline, and peramine. We detected *Epichloë* infections in six seed mixtures, and four contained vertebrate and insect toxic alkaloids typical for *Epichloë festucae* var. *lolii* infecting *Lolium perenne*. As *Epichloë* infected seed mixtures can harm livestock, when infected grasses become dominant in the seeded grasslands, we recommend seed producers to test and communicate *Epichloë* infection status or avoiding *Epichloë* infected seed mixtures.

## 1. Introduction

Infection of crops with fungal endophytes can enhance resistance against environmental stressors, such as drought, herbivores, and pathogens, and therefore have a high potential to increase yield [1,2]. Due to these beneficial traits, some endophytes have been used in seed breeding efforts to symbiotically modify their host plants [3,4]. Economically important fungal endophytes commercially used in grass seed mixtures belong to the genus *Epichloë*, which are naturally distributed in cool season grass species and often produce vertebrate and insect toxic compounds [5,6]. *Epichloë* endophytes are systemic endophytic fungi occurring exclusively in cool season grass species [7]. The anamorphs of *Epichloë* endophytes, formerly known as *Neotyphodium* endophytes, are the most prominent producers of defensive metabolites, which are responsible for livestock intoxications, and of anti-herbivore compounds that are active against insects [8,9,10,11]. The main insect toxic or insect deterring compounds are 1-aminopyrrolizidines, including lolines such as *N*-formylloline and the pyrrolopyrazine, peramine, whereas the ergot alkaloid ergovaline and the indole-diterpene lolitrem B and several of their precursors are toxic to livestock, causing fescue toxicosis in cattle and ryegrass staggers in sheep [10,11,12,13]. Other ergot alkaloids in native grass species can cause, e.g., the drunken horse syndrome [12].

Intoxication of livestock and severe economic losses have been frequently reported from New Zealand, Australia, and the U.S. [9,14] but few cases have been reported from Europe [9,15,16,17]. Severe outbreaks of ryegrass staggers are especially known from New Zealand and Australia [18,19], but have been also documented in Argentina [20], Chile [21], and South Africa [9]. Single cases of ryegrass staggers in Europe have been recorded in Germany, France, the Netherlands, and the United Kingdom [9,15,16,17,22]. Ryegrass staggers are caused by high concentrations of lolitrem B, produced by *Epichloë festucae* var. *lolii* infecting *Lolium perenne* (perennial ryegrass) [9], but lolitrem B can also be produced by the unnamed *Epichloë* sp., found in fine fescues, as well as *Epichloë* sp. FaTG2 in *Festuca arundinacea* (tall fescue) [23,24]. When recognized in a timely manner, the intoxication is reversible [25] and the intoxication risk of humans due to consumption of animal products such as milk is considered to be low, because concentrations of alkaloids in the milk of intoxicated cows are below the toxicity threshold [26]. *L. perenne* can also be infected with the allopolyploid *Epichloë hybrida* (LpTG-2) [27], and the haploid LpTG-3 (such as AR37) that is able to provide resistance against porina larvae (*Wiseana cervinata*) through production of epoxy-janthithrems, which also have tremorgenic effects but not as potent as lolitrem B [28,29,30,31]. Fescue toxicosis is caused by the ergot alkaloid ergovaline, produced predominantly by *Epichloë coenophiala* infecting tall fescue, but has also been observed in some *Epichloë festucae* var. *lolii* isolates. Severe outbreaks of fescue toxicosis are known, particularly from the U.S., with severe economic losses [9,32], whereas reports of fescue toxicosis in Europe are scarce, with one study in France observing ergovaline concentrations in tall fescue above the toxicity threshold [33].

*Epichloë* intoxication events are more common in New Zealand, Australia, and the U.S. compared to Europe because overseas grasslands are often dominated by a single non-native grass species with high endophyte infection rates. For example, the *Epichloë* infection rate of non-native perennial ryegrass is 70% in New Zealand [34] compared to 15% in its native distribution range in Germany [35]. To overcome vertebrate toxicity, *Epichloë* strains have been found and introduced into grass cultivars that are unable to produce the causative toxins [36,37]. One of these is *Epichloë festucae* var. *loii* strain AR1, which still produces alkaloids deterring insects, but not the vertebrate toxic compounds lolitrem B and ergovaline [14,38]. In fact, due to high herbivore pressure, New Zealand seed companies breed perennial ryegrass with vertebrate safe endophytes as an integral part of their cultivars [39].

Perennial ryegrass and tall fescue are important grass species used for temperate grasslands due to their complementary traits, such as the fast initial growth associated with the ryegrasses, and winter hardiness, high production after the second harvest year, and persistence of the fescues [6]. Forage grass seed mixtures are used for developing pastures for agricultural purposes to feed animals [40]. However, turfgrasses are bred for traits that make them useful in lawns used under management practices of varying intensity [41]. Turfgrass mixtures are therefore not meant to feed livestock, but are used on sport fields, airports, parks, yards, or riding arenas.

In other regions of the world, the distribution and toxicity of *Epichloë* endophytes has received much attention [9], but very limited in European grasslands [35,42,43] and almost none in commercially available cultivars and seed mixtures on the European market [43,44]. The aim of this study is to provide an overview on infection rates of grass seeds with *Epichloë* spp. and on the alkaloid content observed in a selection of commercially available forage grass and turfgrass seed mixtures. We want to call attention that sowing of vertebrate toxic endophyte-infected seed mixtures could be a risk for grazing animals in Europe.

## 2. Materials and Methods

### 2.1. Seed Sampling

All seed mixtures are commercially available and were bought from seed breeding companies between March and May 2019. All except one seed mixture contained perennial ryegrass and/or tall fescue. Composition with regard to varieties of perennial ryegrass and tall fescue and the presence of additional grasses is given in Table 1. Samples of the seed mixtures were distributed in a single-blinded and randomized fashion to three laboratories, where they were independently analyzed. After completion of the analyses, the results were combined, and the composition of the seed mixtures was shared. Due to low infection frequencies, additional seeds were analyzed for endophyte infection. The manuscript was drafted by the two first authors without information on seed supplier names to reduce subjective judgements.

### 2.2. Endophyte Detection by Multiplex PCR

Endophyte infection was performed at the Noble Research Institute, LLC, Ardmore, Oklahoma, USA, using a multiplex PCR with endophyte specific primers on DNA extracted from individual seeds randomly selected from each of the seed mixtures, as described previously [35]. Each seed lot was tested twice. In both screens, 22–24 seeds per mixture were tested and the first screen was single blinded. The PCR was performed using only the multiplex 1 primer set [35] with 6 µL instead of 3 µL of DNA, and the number of PCR cycles was increased to 33 cycles as the DNA template concentration was low.

### 2.3. Alkaloid Analyses

Quantitation of the alkaloids ergovaline, peramine, lolitrem B, and paxilline was performed by ultra-high performance liquid chromatography coupled with tandem mass spectrometry (UPLC-MS/MS) using a Waters Acquity UPLC combined with a Quattro Premier triple quadrupole mass spectrometer at the University of Würzburg (Department of Pharmaceutical Biology). Seed samples were ground and prepared, as described previously [45,46]. For extraction, about 9–10 g (25 ml in a Falcon tube) seeds per seed mixture (corresponding to approx. 2000–3000 *L. perenne* seeds) were ground and ground seeds (about 50 mg) were mixed with 250 µl dichloromethane/methanol 1:1 (*v/v*) containing the internal standards lysergic acid diethylamide-D3 (LSD-D3) (125 ng), ergotamine (500 ng), and homoperamine (500 ng). We tested one replicate per seed mixture. After centrifugation, the supernatant was recovered, and the seed pellet was reextracted with 250 µl of dichlormethane/methanol 1:1 (*v/v*). Two aliquots of the supernatant were evaporated under reduced pressure. One sample was reconstituted in water/acetonitrile/formic acid 50:50:0.1 (*v/v/v*) for the analysis of lolitrem B and paxilline, and the other in water/acetonitrile/formic acid 80:20:0.1 (*v/v/v*) for the analysis of peramine and ergovaline.

Chromatographic separation was performed as described previously [47] using 0.1% formic acid in water as mobile phase A and 0.1% formic acid in acetonitrile as mobile phase B. Injection volumes were 10 µl for the analysis of peramine and ergovaline and 5 µl for the analysis of lolitrem B and paxilline. Lolitrem B, ergovaline, and peramine were detected according to [45] using multiple reaction monitoring (MRM). In addition, paxilline was detected using the following specific transitions: m/z 436.3 → 130.1 and 436.3 → 182.1, and LSD-D3 was used as internal standard with the following specific transitions: m/z 327.1 → 208.1 and 327.1 → 226.1. Limit of quantitation (LOQ) for peramine and ergovaline was 0.01 ng on column (0.025 µg/g; 25 ppb), whereas for lolitrem B and paxilline it was 0.05 ng on column (0.125 µg/g; 125 ppb).

In parallel, the levels of lolitrem B and ergovaline were analyzed by the Endophyte Service Laboratory in Corvallis, Oregon, U.S. at Oregon State University, as described previously [48]. For ergovaline analysis, ground seeds were extracted with chloroform, 0.001 M NaOH and ergotamine as the internal standard. After centrifugation, the organic supernatant was applied to Ergosil solid phase extraction (SPE) columns (United Technologies Corporations (UCT), Hartford, Connecticut, U.S.). Columns were washed with a 4:1 acetone chloroform solution, then ergovaline was eluted with methanol. Samples were dried under nitrogen then reconstituted in methanol before analysis via HPLC. For analysis of lolitrem B, ground seeds were extracted with a chloroform/methanol 2:1 (*v/v*) mixture. After centrifugation, the supernatant was recovered and evaporated. The samples were reconstituted in dichloromethane and purified using SPE columns (CUSIL, United Technologies Corporations (UCT), Hartford, CT, USA). Columns were washed with 2.0 mL dichlormethane (DCM), followed by 0.5 mL 4:1 dichlormethane:acetonitrile (DCM:ACN) mixture. Samples were eluted with 3.0 mL of a 4:1 DCM:ACN solution then captured in an amber vial for HPLC analysis. HPLC analysis of ergovaline involved reversed-phase chromatography (Column: Perkin Elmer Brownlee, SPP C18 4.6 × 100 mm, 2.7 µ HPLC column (N9308416) or Phenomenex Kinetex C18 HPLC column; 100 Å, 4.6 × 100 mm, 2.6 µ HPLC column (00D-4426-E0)) at a flow rate of 1.8 ml/min with fluorescence detection at excitation and emission wavelengths of 250 and 420 nm, respectively. The gradient program used 35% acetonitrile and 2 mM ammonium carbonate in water as mobile phase A and acetonitrile as mobile phase B, and a linear gradient running from 99% A to 35% A from 1.0 to 1.6 min. For lolitrem B analysis, the HPLC protocol used normal phase separation (Agilent Zorbax RX-SIL column 4.6 × 250 mm, 5 µ (880975-901); guard column Security Guard column, Silica packing, Upchurch (C-135B)) using dichlormethane/acetonitrile/water (4:1:0.02, (*v/v/v*) as mobile phase, and run at 0.5 ml/min for 15 min with fluorescence detection using excitation and emission wavelengths of 268 and 440 nm, respectively. To generate reference material for use in the calibration curve, ground seed was mixed in large batches at four target concentrations, which was validated using 98% pure ergovaline or purified lolitrem B, respectively. For both ergovaline and lolitrem B, the limit of quantitation (LOQ) in plant samples was 100 ppb (0.1 µg/g).

## 3. Results

In the 24 commercially available seed mixtures included in this study, the vertebrate toxic endophyte alkaloids ergovaline and lolitrem B were quantified by two different laboratories. In addition, paxilline, a vertebrate toxic precursor of lolitrem B, and the insect toxic alkaloid peramine were quantified by one laboratory (Table 2). Four of the seed mixtures (S_10, S_24, S_32, S_33) were found to contain ergovaline and lolitrem B and therefore must have contained *Epichloë* infected seeds. The highest levels of lolitrem B were detected by both laboratories in S_10, a turfgrass seed mixture consisting of three different varieties of perennial ryegrass (Table 1). Paxilline, a precursor of lolitrem B, was also observed in this seed mixture (Table 2). The seed mixtures S_24, S_32 and S_33 all contained different varieties of perennial ryegrass and some additional cool season grasses, but no tall fescue varieties (Table 1). Seed mixture S_33 is also a turfgrass seed mixture, but S_24 and S_32 are forage grass seed mixtures purposed for horse pastures. The insect deterring alkaloid, peramine, was detected in levels over 1000 ppb in S_10, S_24 and S_32 and at lower levels but higher than 10 ppb in S_33, S_13 and S_16 (Table 2).

Detection for the presence of *Epichloë* using PCR identified six seed mixtures (S_10, S_13, S_16, S_24, S_30, S_33) with at least one infected E+ seed (Table 2). In the 18 seed mixtures with no E+ infected seeds, neither ergovaline nor lolitrem B could be detected by both laboratories with the exception of S_32, which contained ergovaline, lolitrem B and peramine. Three of the E+ seed mixtures (S_10, S_24, S_33) were those for which also high levels of ergovaline and lolitrem B could be detected. Two further seed mixtures (S_13, S_16) contained a few seeds with *Epichloë* infections, but did not clearly produce vertebrate toxic compounds, and only low levels of peramine were detected, whereas in seed mixture S_30, no alkaloids could be detected at all. The PCR banding pattern for the single E+ seed in S_30, however, suggests that the E+ seed is likely the result of a meadow fescue (*Festuca pratensis*) sample (Appendix A).

*Epichloë festucae* var. *lolii* that associates with perennial ryegrass is known to produce the alkaloids peramine, lolitrem B, and ergovaline, but each specific endophyte strain may vary with the ability to produce some, or all of these alkaloids. Interestingly, our results show no evidence for *Epichloë* infected tall fescue, as none of the seed mixtures containing tall fescue varieties (S_14, S_16, S_17, S_25, S_29) were shown to contain endophyte-infected seeds or to produce ergovaline, the main vertebrate toxic compound of tall fescue–*Epichloë* associations (Table 1 and Table 2).

## 4. Discussion

We found with analyses of three independent laboratories that four out of 24 commercially available forage grass or turfgrass seed mixtures contained *Epichloë* endophytes, and could produce the vertebrate toxic alkaloids, ergovaline, and lolitrem B. This result is especially surprising for the forage grass seed mixtures (S_24, S_32), as these are used to establish pastures that feed livestock, including sensitive animals such as horses, juveniles, or diseased grazers that are more sensitive to some alkaloids [48]. However, as our study only evaluated the endophyte infection and alkaloid content in the seed mixtures, we cannot conclude the potential alkaloid concentrations and possible toxicity of the pasture after seeding the infected mixtures. Assuming the endophytes are viable at sowing, endophyte-infected plants would be introduced into the environment. *Epichloë* infected agricultural grass species have a selective advantage in hot and dry environmental conditions [49,50,51] and under stressed conditions such as herbivory [52,53,54]. Climate change could result in environmental conditions that are more conducive to survival of infected grasses, which may increase their distribution to a level that endophyte-infected plants could dominate grasslands. Previous studies on grasslands in Germany and Spain showed that single individuals of cool season grass species infected with *Epichloë* can contain alkaloid concentrations above the toxicity thresholds for livestock [35,46,55], which for ergovaline are 300–500 ppb for cattle and horses, and 500–800 ppb for sheep, with lolitrem B concentrations toxic at 1800–2000 ppb [48]. In typical meadows, the infected toxic grasses are diluted by a high diversity of other species and low infection frequencies, so have been of little concern for causing toxicity. However, caution should prevail as our results suggest that the distribution of *Epichloë* infected grasses within Europe could be altered due to planting seed mixtures, and that may inadvertently increase cases of animal toxicity.

The evaluation of the seed mixtures by PCR indicated that the *Epichloë* infected seed mixtures identified in this study did not contain tall fescue, as the PCR markers indicated that the *Epichloë* species present was most likely in the perennial ryegrass *L. perenne*, or possibly in one case, the *Festuca rubra* samples (S_32). Interestingly, the alkaloid biosynthesis marker profile of the seed mixtures analyzed here differed from that of endophyte-infected perennial ryegrass previously sampled on grasslands throughout Germany. The German perennial ryegrass samples typically lacked the marker for the *dmaW* gene and was not able to produce ergovaline [35], whereas seeds in the grass mixtures S_10, S_24, S_32 and S_33 contained this alkaloid. From this, we conclude that the *Epichloë* infected perennial ryegrass seeds in the seed mixtures are likely not from grass native in Germany, but from grass varieties infected with other *Epichloë* isolates that are not safe for vertebrates. The turfgrass seed mixture S_33 contains 5% of the perennial ryegrass variety NEW ORLEANS, which is listed as a top American breeding variety and could be the source of *Epichloë* infected seeds. The low percentage of this variety in the seed mixture could result in sampling variation, which may explain why 8.7% of E+ seeds were detected, and why the alkaloid quantification results differed between the two laboratories. The same supplier that produces S_33 also produces the seed mixtures S_13 and S_16 for which 2.2% and 6.3% E+ seeds were detected, respectively, and very low levels of alkaloids. These results could either indicate low infections with vertebrate friendly *Epichloë* or some seed contamination in the production process, which was not frequent enough to clearly enhance alkaloid concentrations. As these are also turfgrass seed mixtures, it seems implausible that they would lead to intoxication of livestock. However, S_16 is intended for seeding riding arenas and might therefore be grazed by horses, and the *Epichloë* infected grass might spread to horse pastures. We would therefore consider the endophyte presence in this seed mixture could be a matter of concern for horse keepers. For the horse pasture seed mixture S_32, no E+ seeds could be identified, but ergovaline and lolitrem B were detected by both laboratories, as well as high levels of peramine. Interestingly, seed mixture S_28 from the same supplier also intended for horse pastures contains the same varieties of perennial ryegrass, but neither E+ seeds nor alkaloids were detected here (Table 1 and Table 2). Therefore, the potential source of seeds infected with *Epichloë* in the S_32 mixture could also be *Festuca rubra*. Infection rates show high variability (44–92%) in *Festuca rubra* from Spain [56]. However, we have no information on the infection variability that might be seen in different *Festuca rubra* lines used in commercial seed lines. Ergot alkaloids and paxilline have been detected in *Festuca rubra* seeds, but at much lower concentrations than seen in *L. perenne* samples [57].

The data on alkaloid concentrations provided here were determined analyzing the seed mixtures. The actual content of fungal alkaloids in infected *Epichloë* grasses is known to depend on environmental conditions and to be different in the different plant parts [46]. In addition, alkaloids were detectable in lower concentrations in the grasses grown from infected seeds than in the seeds themselves [57]. As *Epichloë* endophyte hyphae accumulate in the developing seeds once the grass enters its reproductive phase, the alkaloid content in seeds in early autumn is especially relevant when grass seed straw or hay is used for feeding animals [58]. As such, while the actual alkaloid concentrations in grasses developing from *Epichloë* infected seed mixtures depend on different factors and may be lower than in the seeds, these grasses contain asexually transmitted *Epichloë* spp. that have the potential to produce considerable levels of vertebrate toxic alkaloids.

In conclusion, our results demonstrate that commercially available turfgrass and forage grass seed mixtures can contain *Epichloë* infected perennial ryegrass varieties producing vertebrate and insect toxic alkaloids. We showed that endophyte infection of commercially available seed mixtures in Europe could have the potential to cause livestock toxicity, especially if left unmonitored. We therefore suggest a number of improvement measures to reduce risks of intoxication of livestock on European pastures due to *Epichloë* infected seed material:(1)When establishing pastures for grazing animals, we suggest avoiding *Epichloë* infected seed mixtures, especially with regard to seed mixtures containing perennial ryegrass varieties.(2)Seed companies could conduct regular tests on *Epichloë* infections of the breeding and seed material and provide detailed information on the exact composition and *Epichloë* infection of the seed mixtures to consumers.(3)The use of *Epichloë* strains incapable of producing the vertebrate toxic compounds lolitrem B and ergovaline could be utilized in European perennial ryegrass breeding, with simultaneous testing the risk of toxicosis.(4)Finally, we would like to call for attention in the EU and other European states to promote research on the neglected risks of intoxications by *Epichloë* infected host grasses on European grasslands.

## Figures and Tables

**Table 1 microorganisms-08-00498-t001:** Composition of grassland seed mixtures, bold indicate seed mixtures with infections of *Epichloë* spp. and detected vertebrate toxic alkaloids. Letters in brackets after the product name indicate if seed mixtures are mainly used as forage grass (F) or turfgrass (T) according to supplier information. Supplier names are in italics.

ID	Perennial Ryegrass Varieties	Tall Fescue Varieties	Other Grass Species	Product
**S_10**	**50% BARCLAY II,** **25% BAREURO,** **25% BARMINTON**	**-**	**-**	**Regenerations-Mischung RPR, *Eurogreen* (T)**
S_11	Unknown variety in unknown percentage	-	*Poa pratensis*, *Festuca pratensis,* *Dactylis glomerata*, *Phleum pratense*, *Festuca rubra*, *Agrostis*	Gräsermischung Weidesaat, *Kräuterwiese* (F)
S_12	25% lawn type, unknown variety 25% pasture type, unknown variety	-	20% *Poa pratensis*, 20% *Phleum pratense*, 10% *Festuca rubra*	Country Horse 2117, *DSV* (F)
S_13	10% CARNAC, 18% CLEOPATRA, 10% DICKENS, 1% EURODIAMOND, 7% DOUBLE 4n, 8% FABIAN 4n, 5% CORSICA, 7% SIRTAKY/SHORTY, 10% ZÜRICH	-	25% *Poa pratensis*	Regeneration Highspeed, *UFA* (T)
S_14	-	100% LIPALMA	-	*Camena Samen* (F)
S_15	33% MATHILDE, 34% WADI, 33% BELIDA	-	-	Elite Gvo, *Rudloff* (F)
S_16	20% EURODIAMOND, 15% SIRTAKY	15% BARCESAR, 35% MEANDRE	15% *Poa pratensis*	Reitbahn, *UFA* (T)
S_17	25% DISCUS	20% LIPALMA	25% *Festuca pratensis*, 20% *Phleum pratense*, 10% *Poa pratensis*	Country Öko 2117, *DSV* (F)
S_18	15% BOYNE, 20% TODDINGTON, 20% INDICUS 1, 15% POLIM, 15% ARUSI, 15% GARBOR	-	-	Profi Nachsaat Gvo, *HaGe Kiel* (F)
S_19	100% KARATOS	-	-	*Camena Samen* (F)
S_20	-	-	7% *Agrostis capillaris*, 3% *Alopecurus pratensis*, 12% *Arrhenatherum elatius*, 10% *Cynosurus cristatus*, 10% *Dactylis glomerata*, 15% *Festuca rubra*, 1% *Holcus lanatus*, 13% *Phleum pratense*, 18% *Poa pratensis*, 1% *Trisetum flavescens*	Heuwiese für Pferde, *Appels Wilde Samen* (F)
S_21	10% KARATOS, 20% KUBUS, 15% TWYMAX	-	25% *Phleum pratense*, 12% *Poa pratensis*, 15% *Festuca rubra*	Pferdeweide 1, *Camena Samen* (F)
S_22	8% PREMIUM	-	18% *Festulolium*, 18% *Phleum pratense*, 15% *Festuca pratensis*	Rotkleegras 91, *Camena Samen* (F)
S_23	100% POLIM	-	-	*Camena Samen* (F)
**S_24**	**12% BELLEVUE**, **20% BOYNE,** **40% STEFANI**	**-**	**18% *Phleum pratense*, 10% *Poa pratensis***	**Pferdeweide Nachsaat, *Raiffeisen* (F)**
S_25	20% IVANA, 10% TIVOLI, 20% SW BIRGER	20% SWAJ	10% *Poa pratensis*, 20% *Phleum pratense*	Pferdegreen Öko PR940, *BSV Saaten* (F)
S_26	100% TWYMAX	-	-	*Camena Samen* (F)
S_27	28% MATHILDE, 23% ALFAN, 13% BELIDA	-	10% *Festuca pratensis*, 5% *Poa pratensis*, 21% *Phleum pratense*	Elite 20, *Rudloff* (F)
S_28	25% MARAVA, 30% BOKSER, 30% WADI	-	15% *Phleum pratense*	Equitana Nachsaat Gvo, *Rudloff* (F)
S_29	10% DOUBLE	45% BARCESAR, 25% DEBUSSY 1	20% *Poa pratensis*	Monaco-Mischung RSM, *Eurogreen* (T)
S_30	11% TREND, 10% TWYMAX, 5% KARATOS	-	10% *Festuca pratensis*, 11% *Festulolium fedoro*, 7% *Dactylis glomerata*, 5% *Poa pratensis*, 5% *Festuca rubra*, 14% *Phleum pratense*	Kräuterweide, *Camena Samen* (F)
S_31	Unknown variety in unknown percentage	-	*Festuca pratensis, Poa pratensis, Poa trivialis, Festuca rubra, Phleum pratense, Alopecurus pratensis, Cynosurus cristatus, Elymus repens*	Pferdeweide-Reparatursaat, *Kräuterwiese* (F)
**S_32**	**15% MARAVA**, **15% BOKSER,** **15% WADI**	**-**	**25% *Phleum pratense*, 20% *Poa pratensis*, 10% *Festuca rubra***	**Equitana Universal, *Rudloff* (F)**
**S_33**	**5% COLUMBINE,** **6% DOUBLE 4 n,** **6% FABIAN 4n,** **12% MERCITWO,** **5% NEW ORLEANS,** **11% SIRTAKY**	**-**	**40% *Poa pratensis*,** **15% *Festuca rubra***	**Primera Highspeed, *UFA* (T)**

**Table 2 microorganisms-08-00498-t002:** Endophyte detection in the Noble Research Institute and quantification of alkaloids at the University of Würzburg and the Oregon State University. Bold numbers/letters indicate that sample was infected/alkaloids were detected. University of Würzburg: one replicate (around 50 mg) of each ground seed mixture was analyzed. Oregon State University: two replicates were analyzed with the exception of samples indicated with ^#^ with only one replicate. nd = not detected; - not analyzed.

	Endophyte Detection Noble Research Institute					Alkaloid Detection [ppb]					
						University of Würzburg				Oregon State University	
	First Screen		Second Screen		Total [%]	Ergovaline	Lolitrem B	Paxilline	Peramine	Ergovaline	Lolitrem B
ID	#Seeds	E+	#Seeds	E+							
**S_10**	**24**	**3**	24	0	**6.3**	**435**	**851**	**1540**	**1899**	**844**	**1688**
S_11	24	0	24	0		nd	nd	nd	nd	<100	<100 ^#^
S_12	24	0	24	0		nd	nd	nd	nd	<100	-
S_13	**22**	**1**	24	0	**2.2**	nd	nd	nd	**10**	<100	<100
S_14	24	0	24	0		nd	nd	nd	**<10**	<100	<100
S_15	24	0	24	0		nd	nd	nd	nd	<100	<100
S_16	**24**	**2**	**24**	**1**	**6.3**	nd	nd	nd	**65**	<100	**104**
S_17	22	0	22	0		nd	nd	nd	**<10**	<100	<100
S_18	24	0	24	0		nd	nd	nd	nd	<100	<100
S_19	24	0	24	0		nd	nd	nd	nd	<100	<100
S_20	24	0	24	0		nd	nd	nd	nd	<100	<100
S_21	22	0	22	0		nd	nd	nd	nd	<100	<100
S_22	24	0	24	0		nd	nd	nd	nd	<100	<100
S_23	24	0	24	0		nd	nd	nd	nd	<100	-
**S_24**	**24**	**3**	24	0	**6.3**	**709**	**240**	nd	**2286**	**425**	**991**
S_25	22	0	22	0		nd	nd	nd	nd	<100	<100 ^#^
S_26	24	0	24	0		nd	nd	nd	nd	<100	<100
S_27	24	0	24	0		nd	nd	nd	nd	<100	<100
S_28	24	0	24	0		nd	nd	nd	nd	<100	-
S_29	22	0	22	0		nd	nd	nd	nd	<100	<100
S_30	24	0	**24**	**1**	**2.1**	nd	nd	nd	nd	<100	<100 ^#^
S_31	24	0	24	0		nd	nd	nd	nd	<100	<100
**S_32**	24	0	24	0		**335**	**494**	nd	**1507**	**287**	**725**
**S_33**	**22**	**2**	**22**	**2**	**8.7**	**7**	nd	nd	**40**	**166**	**372**

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
