# Peer review of "Epichloë Endophyte Infection Rates and Alkaloid Content in Commercially Available Grass Seed Mixtures in Europe"

_microorganisms, 2020, doi:10.3390/microorganisms8040498_

Round 1

Reviewer 1 Report

The paper from Krauss and coauthors perform a survey on grass seeds contamination by 4 toxic metabolites produced by a specific group of endophytic fungi.

In my opinion the paper is well written, it is easy to read it and it also easy to understand the methodoly. Despite the good quality of writing, I'm a little skeptical about the novelty of this work. I'm no an expert of this specific group of fungi, but as a microbiologist I don't understand what the authors want to achieve. They didn't developed any new methods, they didn't looked in the natural environment, they only bought seeds and tested the toxins/fungi presence. Furthermore, this group of fungi is already known, we known that in Europe they are not a problem, but they decided to test anyway the metabolites content and presence/absence of related fungi. It is worth noting that despite they found the fungi in seeds, it is possible that when introduced in the agricultural environment they are replaced by other natural fungi that prevent the accumulation of toxins.

For this reason that the data here presented are not sufficient to be accepted as an article in Microorganisms. I suggest the authors to add at least some further analyses on the plants developed by the seeds and possibly in the natural agricultural environment. 

Author Response

Reply Reviewer 1:

We agree with the reviewer that neither the methods nor the fungal infections are novel per se. However, what is new is the finding that endophyte infection and their production of toxic alkaloids in European seed mixtures could actually constitute a problem and introduces a risk to grazing animals. Part of the seed mixtures were bought on the basis of reports from horse keepers who monitored symptoms of toxicoses. The data presented in the manuscript therefore provide the missing link between the known effect of the toxic alkaloids observed mainly in overseas, and the presence of these endophytes also in European grassland seed mixtures. With this we manage to show that the known hazard of animal intoxication through grazing of endophyte-infected grasses could pose risk also in Europe, as endophyte-infected seeds are sown out on grasslands.

Even if we sowed the infected seed mixtures now, it would take at least three months until we could analyze the grasses (perhaps longer due to pending COVID-19 lockdowns). Furthermore, doing these will in our understanding not provide much novel information, as the actual content of fungal alkaloids is known to depend on environmental conditions and to be different in different parts of the infected grasses (König et al., 2018). In addition, it was already reported that alkaloids were detectable, yet in lower concentrations in the grasses grown from infected seeds than in the seeds themselves (Bauer et al. 2018). The Epichloë endophytes predominantly producing toxic alkaloid are the seed transmitted asexual Epichloë sp., while alkaloid content produced by sexually distributed Epichloë typically is lower (Hume et al. 2016, Leuchtmann et al. 2000). Epichloë endophyte hyphae from asexual, seed transmitted Epichloë species accumulate in the developing seeds once the grass enters its reproductive phase (Pirelli et al., 2016). The high alkaloid content in seeds in September/October is therefore especially relevant, when grass seed straw or hay is used for feeding animals. This literature information has now been added to the discussion to provide a better overview of the known relation between alkaloid content in infected seeds and the resulting grasses (lines 269-278).

We are convinced that the scenario of animal intoxication in the natural environment in which grazing animals are fed with endophyte-infected grasses is sufficiently established. In the text we added another section to the introduction (lines 95-97) and the discussion (281-282)) to clarify the aims of this study to demonstrate that sowing of endophyte-infected seed mixtures corresponds to the intentional and avoidable introduction of a risk for grazing animals, as they get exposed to a known hazard with the consumption of toxic fungal alkaloids. With this we hope to make clear that endophyte infection of commercially available seed mixtures in Europe could constitute a problem. We are convinced that we currently are at an early stage of a possible problem, where awareness by consumers and seed vendors is sufficient to eliminate upcoming intoxications, which is why we think the publication of our data is both timely and important.

Reviewer 2 Report

This study will be of interest to pasture and turf breeders and the community of scientists studying Epichloe and their interactions with their host grasses. The study is well designed and documented, and the appropriate controls and 'blinding' have been implemented. With respect to specific comments:

There should be some comments on the ability of the marker set of PCR primers used to detect and/or differentiate between the suite of endophyte taxa LpTG1, LpTG2 (Hybrid) and LpTG3 (Janthitrem producing) Epichloe endophytes that can be found in perennial ryegrass. A brief mention of Janthitrem producing endophytes in the introduction would be beneficial.

Can the markers distinguish between these different types of endophytes, if so then the absence of LpTG2 and LpTG3 should be noted, if not then then this should be clearly commented on. 

The section described in lines186 to 189 "Interestingly, our results showed no evidence for Epichloë infected tall fescue..." is confusing because of the full stop after the words "tall fescue varieties" Should this refer to tables 1 & 2?

One other comment I have is is 48 seed reps enough given the complex mixtures of species in some of the seed mixtures to confirm endophyte presence by PCR?

The reason I ask this is the levels of alkaloid detection in S 32 and the lack of a positive endophyte seedling in the 48 samples tested. Given the reasonable level of alkaloids detected, the implication from the discussion is that the 10% Festuca rubra in the mix is contributing  the alkaloids, and this combination of host and endophyte hasn't been sampled in the PCR assay. The likelihood of this given that some of the other ~10% Festuca rubra  containing samples (S_12 S_20, S_21) are all alkaloid negative should be covered. How variable are the Festuca rubra lines likely to be in their infection rate.

Another possible addition to the second conclusion point is that European seed companies  look towards New Zealand  seed companies which breed perennial ryegrass with endophytes as an integral  part  of their cultivars. There could also be work  on exploiting and enhancing vertebrate safe Epichloe strains in European perennial ryegrass breeding, as  well as examining the risks of toxicosis

Author Response

Reply Reviewer 2:

We thank the reviewer for confirming that the data presented in the manuscript are of interest to pasture and turf breeders, as well as scientists investigating Epichloë. As mentioned in the reply to reviewer 1, we are convinced that our data manage to show that sowing of endophyte-infected seeds could introduce a risk to grazing animals in Europe due to exposure to toxic fungal alkaloids. With our article we therefore try to raise the awareness and to give recommendations on how to reduce the risk.

There should be some comments on the ability of the marker set of PCR primers used to detect and/or differentiate between the suite of endophyte taxa LpTG1, LpTG2 (Hybrid) and LpTG3 (Janthitrem producing) Epichloe endophytes that can be found in perennial ryegrass. A brief mention of Janthitrem producing endophytes in the introduction would be beneficial.

Reply:

We used a multiplex PCR for the detection of infections as described in Vikuk et al. (2019). To detect infections in the seed mixtures it was sufficient to only use the first primer mix specifically amplifying tefA, a conserved gene encoding Translation elongation factor 1-alpha, perA-T2 encoding Peramine synthetase, lolC, a loline gene marker, dmaW, an ergot alkaloid gene marker, and idtG, an indole-diterpene gene marker (see also Figure S1). With this first primer mix, we were able to detect the initial genes necessary for alkaloid synthesis. A differentiation between loline producing Epichloë (i.e. E. uncinata) and indole-diterpene and ergot alkaloid producing Epichloë (i.e. E. festucae, E. festucae var. lolii) is possible, but it gives us no further information to differentiate between LpTG2 Epichloë endophytes. The jtm locus shares gene clusters with the lolitrem B locus, but there are four genes (jtmD, jtmO, jtm01, jtm02), which are unique to Epichloë spp. producing epoxy-janthitrems, yet we did not test for these genes. Hence, we are not able to differentiate between LpTG1, LpTG2 or LPTG3.We added mention of epoxy-janthitrem producing endophytes in the introduction (lines 64-67).

The section described in lines186 to 189 "Interestingly, our results showed no evidence for Epichloë infected tall fescue..." is confusing because of the full stop after the words "tall fescue varieties" Should this refer to tables 1 & 2?

Reply: In line 187, now line 200, we have corrected the sentence as following: ‘… as none of the seed mixtures containing tall fescue varieties (S_14, S_16, S_17, 187 S_25, S_29) were shown to contain endophyte infected seeds or to produce ergovaline, the main vertebrate toxic compound of tall fescue – Epichloë associations (Tables 1, 2).‘

One other comment I have is is 48 seed reps enough given the complex mixtures of species in some of the seed mixtures to confirm endophyte presence by PCR?

The reason I ask this is the levels of alkaloid detection in S 32 and the lack of a positive endophyte seedling in the 48 samples tested. Given the reasonable level of alkaloids detected, the implication from the discussion is that the 10% Festuca rubra in the mix is contributing  the alkaloids, and this combination of host and endophyte hasn't been sampled in the PCR assay. The likelihood of this given that some of the other ~10% Festuca rubra  containing samples (S_12 S_20, S_21) are all alkaloid negative should be covered. How variable are the Festuca rubra lines likely to be in their infection rate.

Reply: We agree that the analysis of 48 seeds provides only a snapshot of the rate of endophyte infection in complex seed mixtures. However, from this random sampling we can conclude positive statements of endophyte detection, but not the rate of infection of individual grasses in complex mixtures. There is always a chance that some seeds were undersampled. Based on the non-identification of endophyte infection in S_32 we would therefore not postulate a new host-endophyte association.

Infection rates show high variablity (44-92%) in Festuca rubra from Spain vary from (Zabalgogeazcoa et al. 1999). But we have no information on the infection variability that might be seen in different Festuca rubra lines used in commercial seed lines. Ergot alkaloids and paxilline have been detected in Fetuca rubra seeds, but at much lower concentrations than seen in L. perenne samples (Bauer et al. 2018). We added this information in the discussion (lines 265-268).

Another possible addition to the second conclusion point is that European seed companies look towards New Zealand seed companies which breed perennial ryegrass with endophytes as an integral part of their cultivars. There could also be work on exploiting and enhancing vertebrate safe Epichloe strains in European perennial ryegrass breeding, as well as examining the risks of toxicosis

Reply:

We have extended the corresponding section in the Introduction from line 80 on with the following information: ‘In fact, due to high herbivore pressure, New Zealand seed companies breed perennial ryegrass with vertebrate safe endophytes as an integral part of their cultivars.’

We also interchanged the second and third conclusion point and changed the formerly second conclusion point (now third) from line 290 on to: ‘The use of Epichloë strains incapable of producing the vertebrate toxic compounds lolitrem B and ergovaline could be utilized in Eureopean perennial ryegrass breeding, with simultaneous testing the risk of toxicosis.`

Round 2

Reviewer 1 Report

I carefully checked the improvement of the revised version proposed by the authors. In my opinion the new version was not implemented as requested. Only few introduction or discussion paragraphs were added. I still believe that this work miss a connection with the real environment and therefore it remains a survey of fungi through PCR technique.

I understand that experiments require time (especially during this COVID pandemic) but I still believe that they are strictly necessary to publish the present paper in this journal. Otherwise, in my feelings, I would recommend to submit this paper on another journal in which PCR survey with some chemical analyses are commonly accepted.